# Numerical Investigation of Water Inflow Characteristics in a Deep-Buried Tunnel Crossing Two Overlapped Intersecting Faults

Jing Wu [1,*] , Yani Lu [1,*], Li Wu [2], Yanhua Han [1] and Miao Sun [2]

1 Faculty of Civil Engineering, Hubei Engineering University, Xiaogan 432000, China
2 Engineering Research Center of Rock-Soil Drilling & Excavation and Protection, Ministry of Education, China University of Geosciences, Wuhan 430074, China
* Correspondence: wujing@hbeu.edu.cn (J.W.); lyn2016@hbeu.edu.cn (Y.L.)

**Abstract:** Because fault core zones and damage zones overlap, when a tunnel crosses the intersecting faults the groundwater flow characteristics of the tunnel-surrounding rock will be different compared to that from a single fault. By using the theory of "Three-district zoning of faults", an improved Darcy–Brinkman numerical model for a tunnel crossing the intersecting faults was established in this work. Based on the relative vertical positions between the tunnel axis and the intersection center of faults, the underground water seepage field was analyzed at steady-state by solving the improved Darcy–Brinkman equation for the host rock zone and the fault zone. The simulation results show that the flow field around the tunnel is almost unaffected by the relative positions but is mainly dependent on the relative heights. Specifically, the relative position variation of the fault intersection to the tunnel axis has little effect on the pore pressure. In terms of flow velocity, regardless of the relative positions of the fault intersection and the tunnel, the maximum value of flow velocity almost occurs near the bottom of the tunnel excavation face and consistently displays high values within a small distance ahead of the excavation face, and then decreases quickly as the distance increases. Furthermore, the flow velocity changes minimally in the host rock. It will likely encounter the maximum water inflow rate when the tunnel excavation face passes through the intersection. The numerical simulation results can provide a practical reference for predicting water inflow into deep-buried tunnels passing through overlapped intersecting faults.

**Keywords:** deep-buried tunnel; three-district zoning of faults; intersecting faults; water inflow analysis





## 1. Introduction

With the rapid development of China's economy, underground projects, such as tunnel constructions in the southwestern mountainous areas, are faced with complex geological conditions and frequent geohazards [1–4]. Water inrush disasters have become a significant geohazard during the construction progress of deep-buried tunnels. Many major safety accidents are caused by water inrush without proper prediction and mitigation. Therefore, studying water inrushes caused by adverse geological structures has become the focus of underground engineering research [5–10]. The key to the frequent occurrence of accidents is the lack of understanding of water inrush mechanisms, effective forecasting methods, and mitigation measures [3–7]. It is essential to investigate the evolution law of water inflow into the tunnel and construct practical forecasting and prevention countermeasures.

In recent years, many studies on water inrush prevention in tunnel engineering have focused on water gushing in mines [11–15]. Predicting water inrush caused by fault damage zones has gradually become a focus. The integrity of the rock and soil in damaged zones is poor, and the structure is often broken and loosened. When tunnel excavations pass through damage zones with water-rich faults, under the action of groundwater and construction disturbances, the rock and soil are prone to disintegration and modification, forming water-conducting channels. Improper operation handling under such conditions can result in

water and mud inrush disasters [16–18]. Through theoretical analysis, physical model testing, and numerical simulation, scholars have systematically studied the mechanisms of water inrush when tunnels cross faults. For example, Wu et al. [19] simulated the water inrush process by solving the Darcy–Brinkman equation for the host rock and the fault zone and examined the impacts of the angle between the tunneling direction and the fault and the relative position from the tunnel face to the fault on the evolution of pore pressure and flow velocity near the tunnel face. Jeon et al. [20] conducted physical model experiments to investigate the influence of faults, the presence of weak planes, and grouting on rock stability and found that displacement changes and shear deformation increase significantly at the fault and weak planes. Zheng et al. [21] established a numerical model of water inrush into tunnels induced by filled fault zones, and obtained the distribution of unstable zones over time and proposed an "activation coefficient" to characterize the evolution process of fault damage zones.

In engineering practice, it is often encountered that two faults with different orientations form an intersecting fault system [22]. At the intersection of the two faults, fault cores and damage zones overlap. Under the combined influence of two overlapped intersecting faults, groundwater inflow characteristics are different from those induced by a single fault. It is important to learn the mechanism and evolution law of water inrush under the impact of intersecting faults. In this work, by using the theory of "Three-district Zoning of faults", we established an overlapped intersecting fault model and simulated the the laminar-turbulent flow in highly heterogeneous fault formations By solving the Darcy equation for the groundwater flow in the host rock zone and the improved Brinkman equation for the groundwater flow in the fault zone, we replicated the water inrush into the tunnel affected by two intersecting faults and explored the groundwater seepage behavior while the tunnel excavation passed through two overlapped intersecting faults. Finally, we analyzed how the relative intersection position and the relative height value affected the seepage characteristics in the tunnel-surrounding rock and the water inflow rate into a tunnel.

## 2. Numerical Investigations

### 2.1. Numerical Calculation Model

On the profile perpendicular to the strike, the anatomical structure of the fault can usually be concluded with three different zones, including the core zone, the damage zone, and the host rock zone, which can be called "Three-district zoning of faults". In other words, when the tunnel excavation face passes through a fault geological structure, it will sequentially pass through the host rock zone, the damage zone, the core zone, the damage zone again, and the host rock zone. In this kind of fault structure zonation, the host rock zone has the same physical and mechanical characteristics as the rocks in situ. Various fault rocks and associated fractures are developed in the fault structure. The rock mass in the fault structure is fractured rock mass, with a random fabric mainly including fault gouge, fault breccia, cataclasite, and mylonite, with large permeability and water conductivity. The damage zone is a transitional zone between the host rock zone and the core zone of fault, which has not been seriously influenced by the geological conformation movement or the dynamic tunnel excavation progress. The range and the distribution of cracks and weak structural planes in the damaged zone are affected by complicated factors, for examples the original geostress, the classification and scale of faults, stratum lithology, and the depth of tunnel. These cracks in the rock mass of the fault usually represent high permeability because they are not filled with minerals.

In this work, the base width of the tunnel cross-section is 3.0 m and the diameter of the upper semicircular area is 3.9 m. The dimensions in the X, Y, and Z directions of the numerical investigation model are 160 m × 100 m × 100 m, where the *x*-axis is the excavation orientation of the tunnel central axis. The average tunnel depth is 350 m and the groundwater buried depth is 50 m. The dip angles of the two faults are both 45°, with the fault planes perpendicular to each other. The strike of the two faults is consistent, which is perpendicular to the strike of the tunnel. In this work, the host rock represents

limestone and dolomite. In the numerical model, the permeability value of the host rock zone is $10^{-16}$ m$^2$, the maximum permeability value of the fault core zone is $10^{-11}$ m$^2$, and the variation in permeability in damage zones obeys the Gaussian function. The rock permeability in the fault intersection area follows the principle of superposition. As displayed in Figure 1, zone A is the surrounding host rock zone, zone B is the fault damage zone as a transition zone, zone C is the fault core zone, and zone *D* is the overlapped damage zone of the faults. The absolute widths of the core zone C, the damage zone B, and overlapped damage zone *D* are 3 m, 20 m, and 20 m, respectively. The porosity value of the fault core zone and the fault damage zone is 0.5. Based on the simulation model of these intersecting faults, we explored the influence of the vertical relative positions and distance between the tunnel axis and the fault's intersecting center on the seepage characteristics of groundwater and water inflow into the tunnel. In this study, we set the tunnel excavation face and the fault's intersection center on the XY plane. Specifically, the calculation conditions are that while the tunnel excavation face enters the center of the model, the center of the fault intersection is located 40 m and 20 m directly above the tunnel excavation face (Y = 40 and Y = 20), coinciding with the tunnel axis (Y = 0), and 20 m and 40 m directly below the tunnel excavation face (Y = -20 and Y = -40), as shown in Table 1.

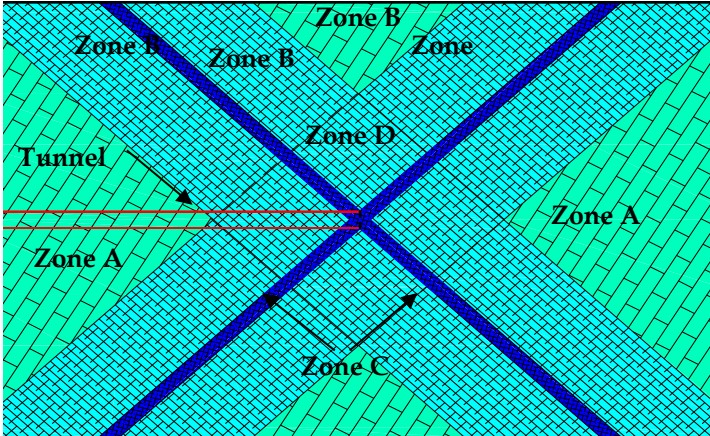

**Figure 1.** Numerical simulation model of a tunnel crossing two overlapped intersecting faults.

**Table 1.** Simulation cases for a tunnel crossing two intersecting faults.

| Factor | Classification | | | | |
|---|---|---|---|---|---|
| The location of intersecting faults (m) | Y = 40 | Y = 20 | Y = 0 | Y = −20 | Y = −40 |

### 2.2. Numerical Simulations

In this work, we utilize COMSOL Multiphysics to solve the Darcy equation for the linear groundwater flow in the host rock zone and the improved Brinkman equation for the nonlinear groundwater flow in the fault zone. The groundwater flow behavior and the water inflow into the tunnel passing through two intersecting faults were simulated. Assuming that the rock mass is a porous medium within the host rock zone, the groundwater flow follows the Darcy flow as in Equation (1).

$$\begin{cases} \nabla \cdot (\rho_w \cdot u) = Q_m \\ u = -\left(\frac{k}{\mu_w}\right)(\nabla p + \rho g \nabla H) \end{cases} \tag{1}$$

where $u$ is the flow velocity (*FV*), $\rho_w$ is groundwater density, $Q_m$ is flow rate, $k$ is permeability, $p$ is the pore pressure (*PP*), $\mu_w$ is the dynamic viscosity of groundwater, $H$ is groundwater head height.

The upper, left, right, and back boundaries of the model are the influent boundary, that is, the source recharge conditions of groundwater seepage. We assume that sufficient rainwater recharge conditions exist, and the inlet boundary is set at pressure boundary conditions. The host rock section has a pressure boundary condition, which must meet the condition as in Equation (2).

$$p = p_0 = \rho_w g(300 - Z) \tag{2}$$

where $Z$ at the tunnel axis plane is set to 0. Additionally, due to contact with the atmosphere, *PP* on the excavation face and the excavated tunnel wall 1.0 m rearwards can be set to 0. Except for this, the tunnel excavation perimeter, the bottom boundary, and the front boundary of the model are all set as impervious boundaries as shown in Equation (3).

$$\vec{\mathrm{n}} \cdot \frac{k}{\mu_w}(\nabla p + \rho g \nabla H) = 0 \tag{3}$$

The shear stress of groundwater cannot be neglected in the fault damage zone or the fault core zone because of the relatively significant energy that is consumed by the shear effect. As a result, the groundwater seepage progress can be considered as a nonlinear turbulent flow with a big flow velocity value in the fault zones. The Brinkman equation expresses this groundwater seepage behavior as Equation (4).

$$\left. \begin{array}{l} \nabla \cdot \left\{ \frac{\eta_w}{\varepsilon_p} \left[ \nabla u + (\nabla u)^T \right] - pI \right\} - \frac{\eta_w}{k} \cdot u + \mathrm{F} = 0 \\ \rho_w \nabla \cdot (u) = Q_m \end{array} \right\} \tag{4}$$

Field water pressure test was carried out in the vicinity of the F61 fault in the diversion tunnel. The dip angle of the F61 fault is 42°, the width of the core zone is 3 m, and the damage zone is 20 m. The permeability of the host rock zone is $k_r = 1 \times 10^{-16}$ m$^2$, and the permeability coefficients of the core zone and damage zone are calculated according to the water compression test. According to the results of the water compression test, the influence relationship between permeability $k$ and distance $x$ from the fault center was studied, and the relationship between $x$-$k$ was fitted by Gaussian function. Where $I$ is the unit matrix, $\varepsilon_p$ is a void ratio, $F$ is the volume force, and $k$ is the permeability of the fault surrounding rock which can be expressed as Equation (5).

$$\begin{cases} k = k_f \cdot e^{\frac{(\ln k_f - \ln k_r)x^2}{\left(\frac{d_1}{2} + d_2\right)^2}}, |x| \le \frac{d_1}{2} + d_2 \\ k = k_f, |x| \ge \frac{d_1}{2} + d_2 \end{cases} \tag{5}$$

where $k_f$ is the constant permeability value in the core zone of the fault, $k_r$ is the constant permeability value of the host rock zone, $d_1$ and $d_2$ are the absolute width of the core zone and the damage zone, respectively, and $x$ is the distance from the fault center.

Combining the above equations yields a mathematical calculation model of nonlinear turbulent flow in fault zones based on the improved Brinkman equation. The permeability change can be achieved in the software COMSOL.

The transition of various groundwater flows in the fault zone and the host rock zone obeys certain boundary conditions. Using the condition of fluid mass balance and fluid pressure balance, the numerical simulation of water flow in the fault zone can be realized by combining Darcy flow and the improved Brinkman flow. Specifically, the host rock zone follows Darcy flow, and the fault zone follows the improved Brinkman flow. The continuous *PP* and *FV* conditions are met at the interface of each two adjacent regions, which can be expressed as Equation (6).

$$\begin{cases} p_D(B_i) = p_B(B_i) \\ u_D(B_i) = u_B(B_i) \end{cases}, (i = 1, 2) \tag{6}$$

In combination with the above Equations from (1) to (6), by using the mass conservation and pressure balance conditions, we obtain the turbulent seepage mathematical model in the fault zone to replicate the laminar flow simulation in the surrounding rock.

## 3. Numerical Simulations When Tunnel Crosses Two Intersecting Faults

### 3.1. Tunnel Is 40 m Beneath the Center of the Two Intersecting Faults

Figure 2 reveals *PP* contours and *FV* contours on various sections ($XZ_{Y = 40}$, $XY_{Z = 0}$, $YZ_{X = 0}$) while the tunnel is excavated to 40 m beneath these two intersecting faults (Y = 40).

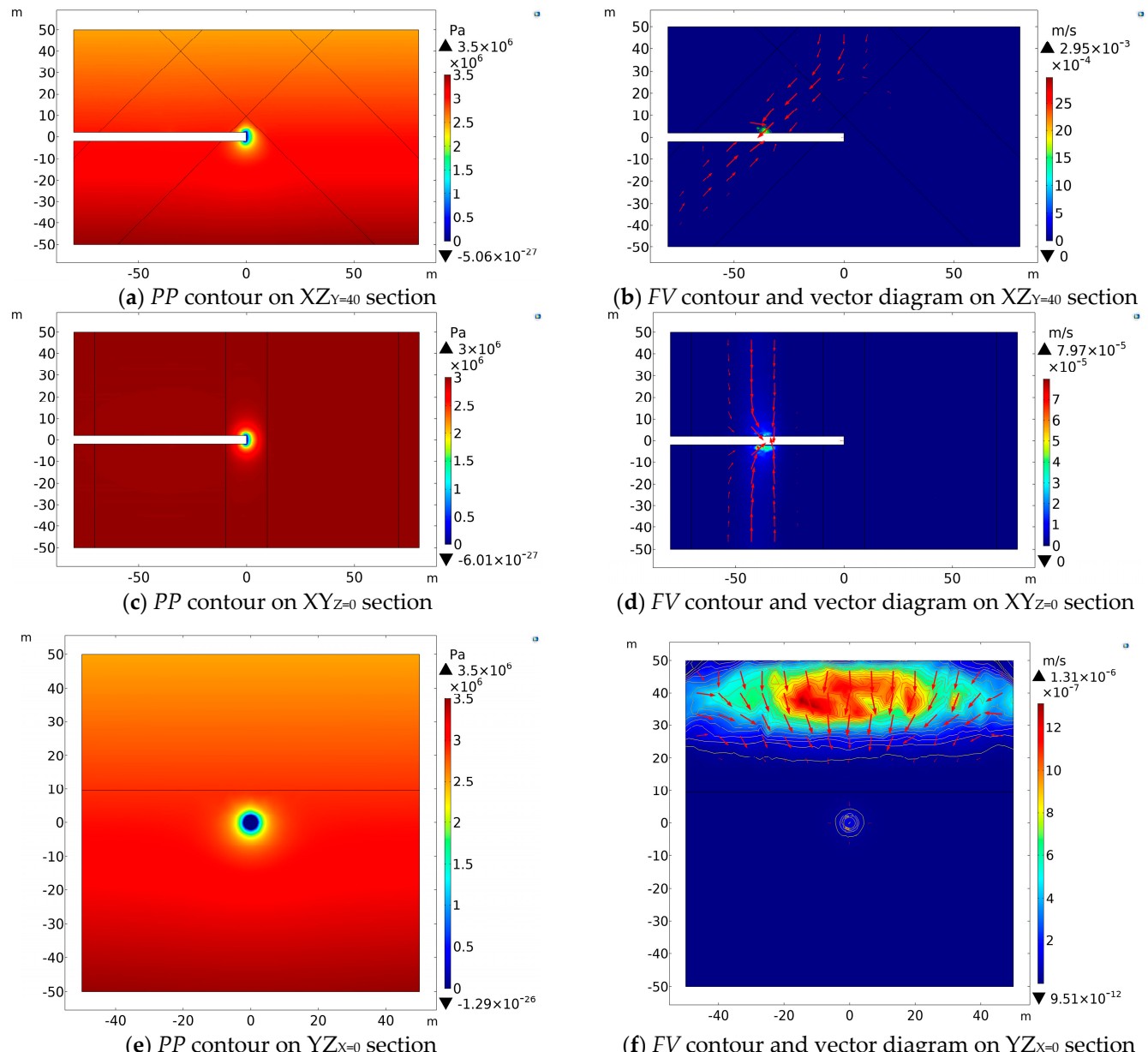

(**a**) *PP* contour on $XZ_{Y=40}$ section

(**b**) *FV* contour and vector diagram on $XZ_{Y=40}$ section

(**c**) *PP* contour on $XY_{Z=0}$ section

(**d**) *FV* contour and vector diagram on $XY_{Z=0}$ section

(**e**) *PP* contour on $YZ_{X=0}$ section

(**f**) *FV* contour and vector diagram on $YZ_{X=0}$ section

**Figure 2.** Simulation results for the case Y = 40 m.

The *PP* contours in Figure 2a,c,e show that a low-pressure region appears at the excavation face and on the tunnel wall within 1.0 m rearwards and the *PP* increases outward with the increases in distance. On the cross-section of $XZ_{Y = 0}$, *PP* contours are distributed elliptically symmetrically along the *x*-axis, and the ellipse's major axis is parallel to the *z*-axis. In the vertical downward direction, the *PP* value gradually increases with the burial depth of the tunnel in a circular or semicircular distribution nearby the tunnel face, as can

be seen in the cross-section of $YZ_{X=0}$. The *FV* contours reveal that the groundwater flow direction is from the fault intersection zone and the damage zone to the tunnel excavation face, and the FV value is small with a maximum of $2.95 \times 10^{-3}$ m/s. The *FV* contours are symmetrically distributed with respect to the *x*-axis on the cross section of $XZ_{Y=40}$ and the *y*-axis on the cross section of $YZ_{X=0}$. While the tunnel excavation face arrived at 40 m directly below the overlapped intersection center, the *PP* near the excavation face and the tunnel wall about 1 m behind dissipated, forming a low-pressure zone. Therefore, groundwater is more likely to seep into the vicinity of the tunnel face. The permeability of the fault core zone and fault damage zone is much bigger than that of the host rock zone on either side, so the direction of the groundwater seepage is from the fault intersection zone and damage zone to the tunnel.

Five measuring lines were placed 50 m in front of the excavation face to monitor and explore the variation law of *PP* and *FV* as listed in Table 2 and Figure 3.

**Table 2.** Information and location of measuring lines and points.

| Number | Location of Measuring Line | Number of Points |
|--------|---------------------------|------------------|
| 1 | X = 0~50 m, Y = 3.90 m | 100 |
| 2 | X = 0~50 m, Y = 1.95 m | 100 |
| 3 | X = 0~50 m, Y = 0.00 m | 100 |
| 4 | X = 0~50 m, Y = −1.95 m | 100 |
| 5 | X = 0~50 m, Y = −3.90 m | 100 |

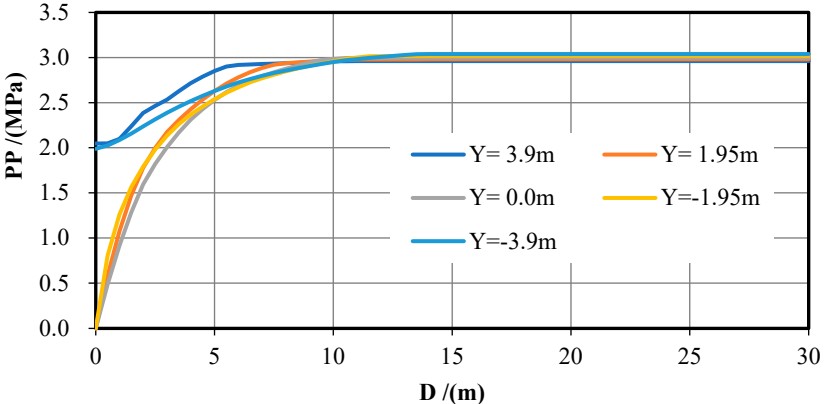

(**a**) *PP* as a function of distance away from the tunnel face (*D*).

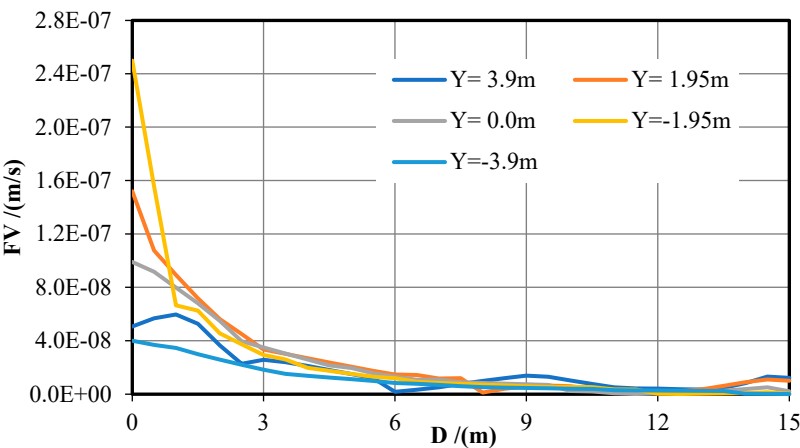

(**b**) *FV* as a function of distance away from the tunnel face (*D*).

**Figure 3.** Pore pressure and flow velocity within 30 m ahead of the tunnel face as a function of distance away from the tunnel face (Y = 40 m).

The *PP* variation law on each measuring line about 30–50 m in front of the excavation face is relatively stable and approaches a maximum of 3.0 MPa. Therefore, we analyzed pore pressure within 30 m in front of the excavation face, and the results are shown in Figure 3a. Flow velocity does not change significantly in the 15–50 m range and is close to zero. Figure 3b shows flow velocity curves 15 m in front of the working face.

When Y = 3.9 m, the *PP* value is 2.05 MPa at the excavation face and improves quickly within 5 m ahead, then keeps increasing slowly deep into the formation, and stabilizes at approximately 3 MPa. When Y = −3.9 m, the *PP* value is 1.99 MPa at the excavation face and improves gradually within 10 m ahead and eventually stabilizes at about 3 MPa. When Y = 1.95 m, 0, and −1.95 m, the *PP* value at the excavation face is 0 and improves quickly within 5 m ahead and finally stabilizes at about 3 MPa. Within 5 m ahead of the excavation face, the values of *PP* on sections of Y = ±3.9 m are significantly bigger than that on sections of Y = ±1.95 m and 0. In summary, when the tunnel excavation face arrives at 40 m directly below the fault intersection center, *PP* in the short distance ahead improves quickly, and after that it grows tardily. Along the *y*-axis, *PP* in the range of −1.95 m ≤ Y≤1.95 m is significantly low near the tunnel excavation face and decreases with an increasing Y away from the excavation face.

As shown in Figure 3b, on the Y = −3.9 m and Y = −3.9 m sections, *FV* varies insignificantly near the tunnel excavation face. The maximum value of *FV* at *D* = 1 m and *D* = 0 is $5.97 \times 10^{-8}$ m/s and $4.01 \times 10^{-8}$ m/s each. On sections of Y = 1.95 m, 0, −1.95 m, *FV* declines quickly and slowly within 3 m ahead of the excavation face. The maximum value of *FV* is $1.52 \times 10^{-7}$ m/s, $9.91 \times 10^{-8}$ m/s, and $2.50 \times 10^{-7}$ m/s, respectively. The velocity magnitude order near the tunnel face is $U_{Y = -1.95} > U_{Y = 1.95} > U_{Y = 0}$. Overall, the *FV* peak occurs at the lower part of the excavation face. The value of *FV* is higher over a small distance in front of the excavation face but then falls off rapidly, after which it changes smoothly in the host rock. *FV* near the excavation face is apparently higher than that outside away from the excavation face, and it increases with increasing Y.

In terms of water inflow, we acquire the groundwater inflow rate of 0.0054 m$^3$/h at the excavation face and the total groundwater inflow rate of 0.015 m$^3$/h by integrating the *FV* over the area.

### 3.2. Tunnel Is 20 m Beneath the Center of Cross Faults

When the tunnel is excavated to 20 m beneath these two intersecting fault centers (Y = 20 m), the contours of *PP* and *FV* on various sections (XZ$_{Y = 20}$, XY$_{Z = 0}$, YZ$_{X = 0}$) are shown in Figure 4.

The *PP* contours in Figure 4a,c,e show that a low-pressure region appears at the excavation face and the tunnel wall within 1.0 m rearwards. Specifically, *PP* value is zero at the excavation face, which then increases with the distance away from the excavation face *D*. On the XZ$_{Y = 20}$ section, the *PP* contours are symmetrically circular along with the *x*-axis. The pore pressure distributes semi-circularly near the excavation face on the vertical section and gradually increases with the increase in the tunnel burial depth. From the *FV* contours, we find that water flows to the tunnel from the intersection zone and the damage zone of faults above the tunnel. As the fault intersection center is closer to the tunnel, the water flow velocity increases, with a maximum of $1.25 \times 10^{-3}$ m/s. On the XZ$_{Y = 20}$ section, the *FV* contour appears approximately symmetrically distributed for the *x*-axis. Water flows along the fault into the tunnel excavation face. On the YZ$_{X = 0}$ section, the *FV* contour is symmetrically distributed for the *y*-axis. The value of *FV* above the tunnel vault is expressively more significant than below the tunnel floor.

In summary, since the fault's intersection center is closer to the tunnel face, it has a more pronounced influence on the distribution of pore pressure in the tunnel-surrounding rock. The pore pressure dissipates faster along the intersection of the faults than perpendicular to the intersection direction. Therefore, the pore pressure distribution changes from elliptical to circular on the horizontal section near the excavation face. The groundwater flows to the tunnel from the overlapped intersection zone and the damage zone of faults above the

tunnel, and the value of *FV* increases exponentially. The water velocity above the tunnel vault is expressively bigger than below the tunnel floor. Five survey lines are selected within 50 m in front of the tunnel excavation face. Table 2 lists the locations of the survey lines. The curves of *PP* and *FV* are shown in Figure 5.

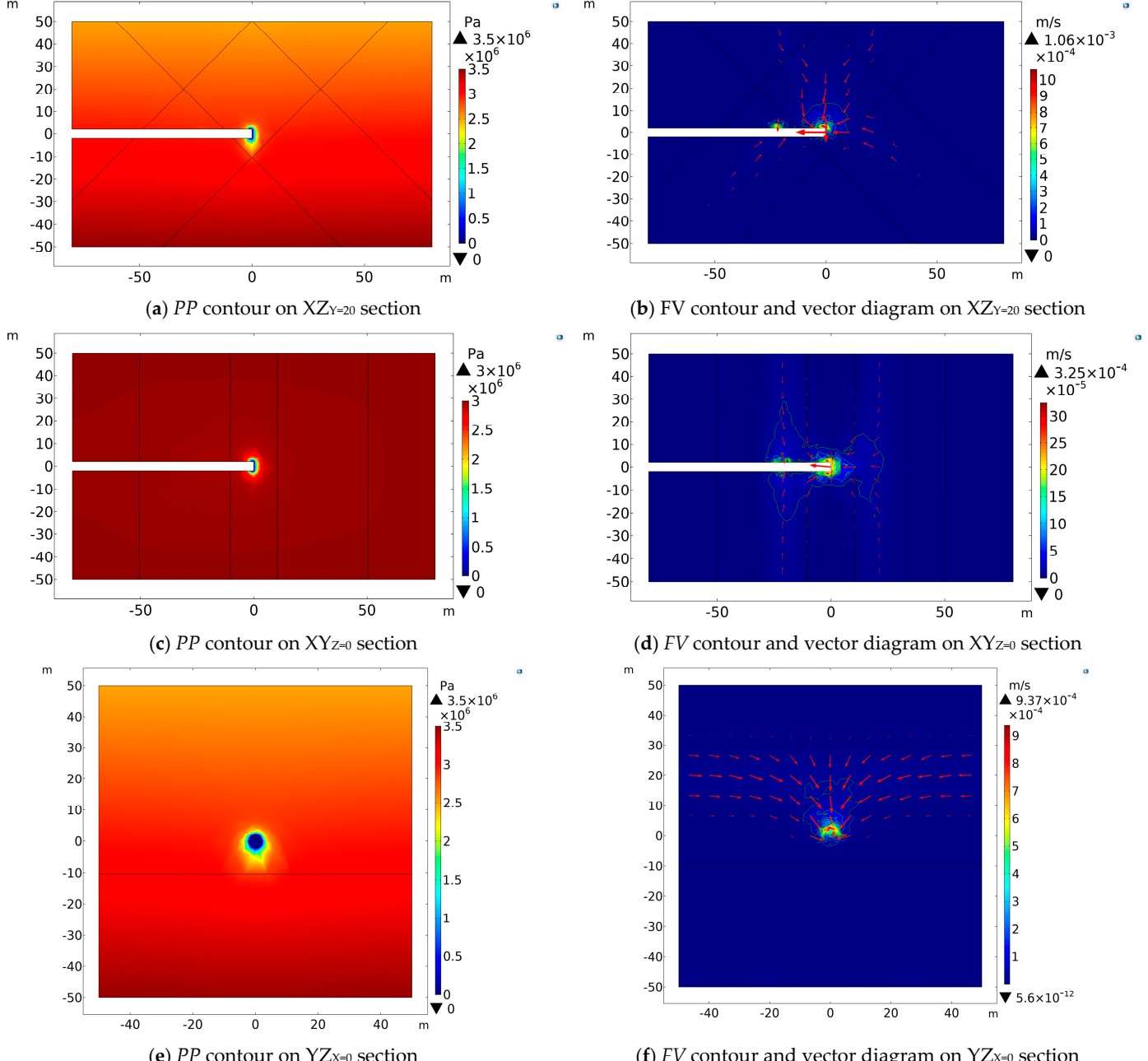

(**a**) *PP* contour on XZ$_{Y=20}$ section

(**b**) FV contour and vector diagram on XZ$_{Y=20}$ section

(**c**) *PP* contour on XY$_{Z=0}$ section

(**d**) *FV* contour and vector diagram on XY$_{Z=0}$ section

(**e**) *PP* contour on YZ$_{X=0}$ section

(**f**) *FV* contour and vector diagram on YZ$_{X=0}$ section

**Figure 4.** Simulation results for the case Y = 20 m.

Figure 5a shows that, on the Y = 3.9 m section, the PP value at the tunnel face is 2.50 MPa, which then gradually increases within 10 m in front of the excavation face and eventually stabilizes at 3.0 MPa. On the Y = −3.9 m section, the value of *PP* at the excavation face is 1.73 MPa, which then gradually increases within 5 m in front and eventually stabilizes at 3 MPa. On the Y = ±1.95 m and Y = 0 section, the value of *PP* at the excavation face is 0, which increases rapidly within 5 m in front, then eventually stabilizes at 3 MPa. The value of *PP* on the Y = ±3.9 m section is apparently more significant than that on the Y = ±1.95 m and 0 sections within 5 m in front of the excavation face. Figure 5b reveals that, on the

Y = 3.9 m and 1.95 m sections, the maximum value of *FV* at $D = 1$ m is $2.57 \times 10^{-4}$ m/s and $6.60 \times 10^{-4}$ m/s, respectively. Flow velocities gradually decrease within $1 < D \leq 9$ m range of the excavation face until reaching zero as $D > 9$ m. When Y = 0, the maximum value of *FV* occurs at the excavation face, which is $2.12 \times 10^{-4}$ m/s. Flow velocity is gradually reduced within $0 < D \leq 9$ m range of the excavation face until zero at $D > 9$ m. When Y = −3.9 m and −1.95 m, the velocities are low and overall change little.

In terms of water inflow, we acquire the water inflow rate of 12.45 m$^3$/h at the excavation face and the total water inflow rate of 35.67 m$^3$/h by integrating the *FV* over the area.

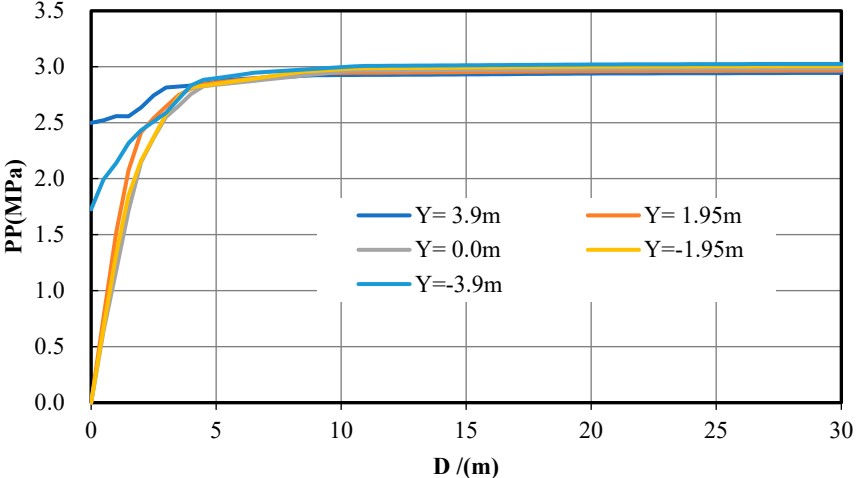

(**a**) *PP* as a function of distance away from the tunnel face (*D*)

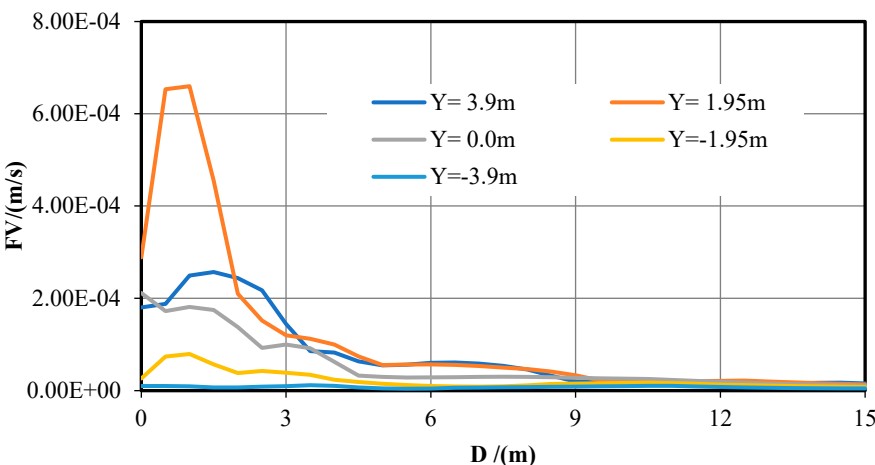

(**b**) *FV* as a function of distance away from the tunnel face (*D*)

**Figure 5.** Pore pressure and flow velocity within 30 m ahead of the tunnel face as a function of distance away from the tunnel face (Y = 20 m).

### 3.3. Tunnel Is at the Center of the Fault Intersection

Figure 6 reveals *PP* contours and *FV* contours on various sections (XZ$_{Y = 0}$, XY$_{Z = 0}$, YZ$_{X = 0}$) while the tunnel excavation face is in the center of the two intersecting faults (Y = 0).

The *PP* contours in Figure 2a,c,e show that a low-pressure region appears at the excavation face and the tunnel wall within 1.0 m rearwards and *PP* increases outward as the distance increases. Specifically, the value of *PP* is 0 at the excavation face and quickly increases outward as the distance increases. On the XZ$_{Y = 0}$ section, *PP* contour appears elliptical symmetrically for the *x*-axis, with the ellipse's major axis along with the *z*-axis. On the vertical profile, the *PP* contour distributes in a funnel shape. The

*FV* contours in Figure 2b,d,f show that groundwater flows mainly from the overlapped intersection zone and the damage zone of faults to the tunnel. The *FV* value is increased with a maximum magnitude of 0.0205 m/s. On the $XZ_{Y=0}$ section, the *FV* contour is symmetrically distributed along the *x*-axis. Groundwater flows from both sides of the tunnel to the inside along the fault intersection.

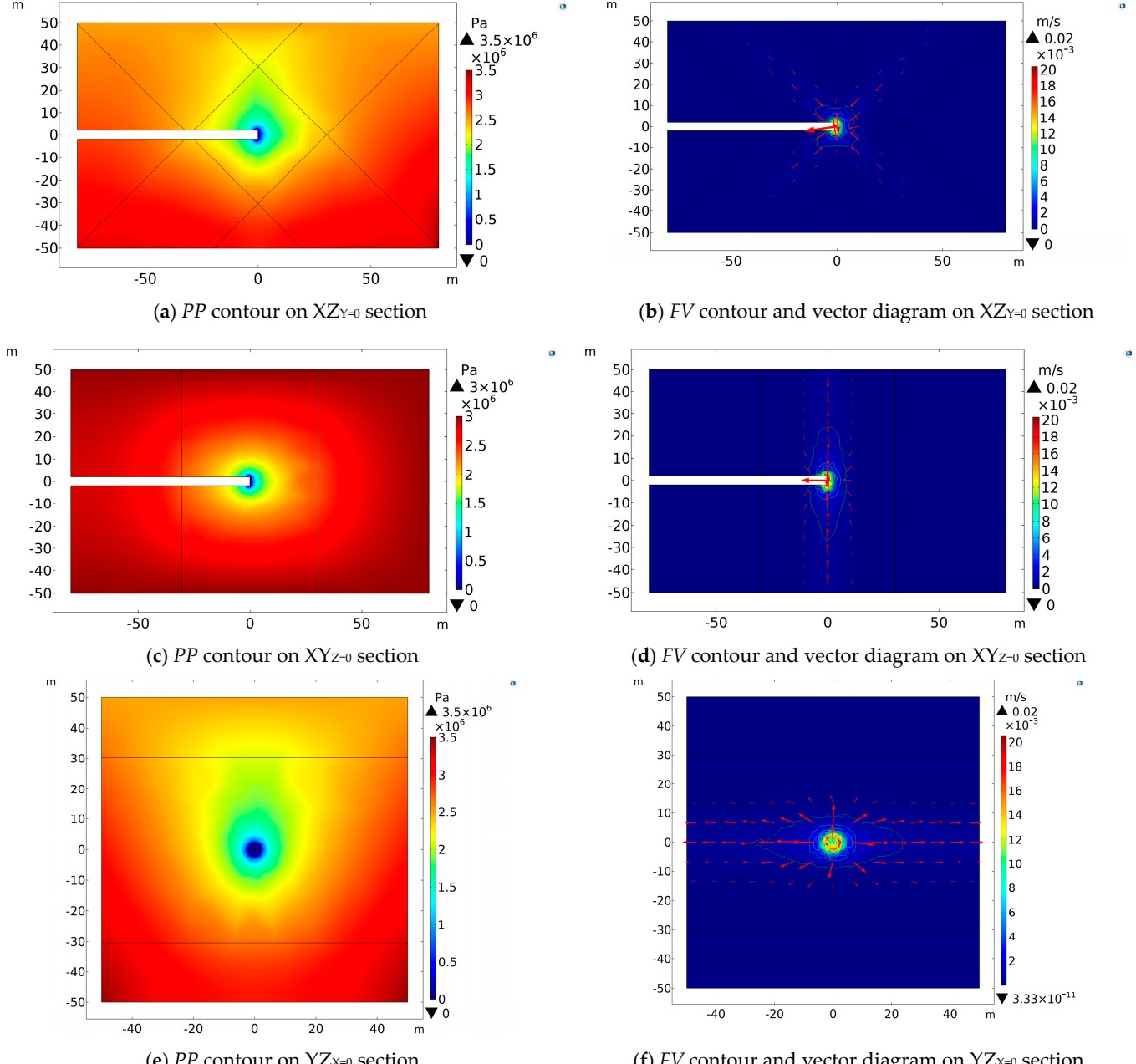

(**a**) *PP* contour on $XZ_{Y=0}$ section

(**b**) *FV* contour and vector diagram on $XZ_{Y=0}$ section

(**c**) *PP* contour on $XY_{Z=0}$ section

(**d**) *FV* contour and vector diagram on $XY_{Z=0}$ section

(**e**) *PP* contour on $YZ_{X=0}$ section

(**f**) *FV* contour and vector diagram on $YZ_{X=0}$ section

**Figure 6.** Simulation results for the case Y = 0 m.

Five measuring lines were placed 50 m in front of the excavation face to monitor and explore the variation law of *PP* and *FV* as listed in Table 2 and Figure 7.

On the Y = 3.9 m and Y = −3.9 m sections, it can be seen in Figure 7(a) that the values of *PP* at the excavation face are 1.12 MPa and 1.05 MPa, respectively. They increase rapidly with the increase in *D* within 5 m ahead, then increase slowly and eventually stabilize at 3.0 MPa. When Y = ±1.95 and Y = 0, the *PP* value at the excavation face is 0 and increases quickly within 5 m ahead and finally stabilizes at about 3 MPa. Within 5 m ahead of the

excavation face, the values of *PP* on sections of Y = ±3.9 m are significantly bigger than that on sections of Y = ±1.95 m and 0, and the *PP* value on the section of Y = −3.9 m is larger than on the section of Y = 3.95 m.

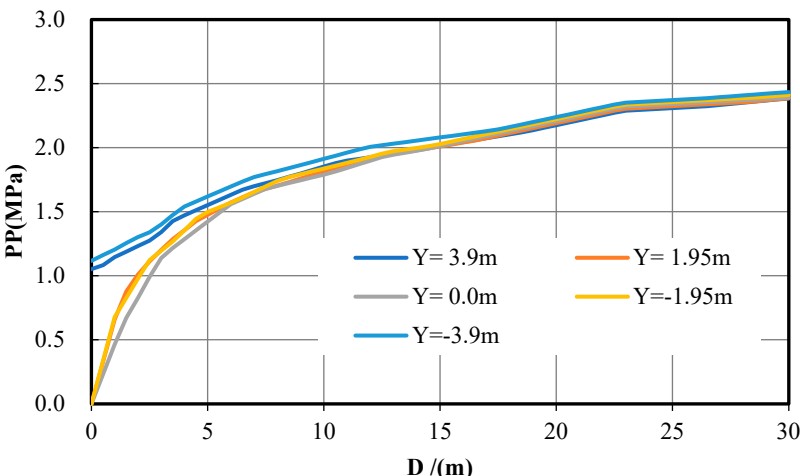

(**a**) *PP* as a function of distance away from the tunnel face (*D*)

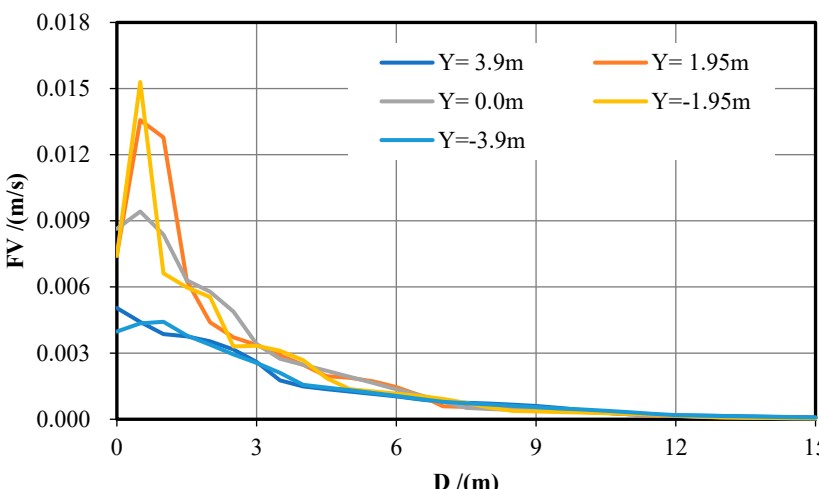

**Figure 7.** Pore pressure and flow velocity within 30 m ahead of the tunnel face as a function of distance away from the tunnel face (Y = 0 m).

Figure 7b shows that, on the Y = 1.95 m, 0, and −1.95 m sections, the maximum values of *FV* appear at *D* = 1 m, which are 0.014 m/s, 0.009 m/s, and 0.015 m/s, respectively. Then, it decreases rapidly within 9 m ahead of the excavation face until reaching zero. When Y = 3.9 m and Y = −3.9 m, the maximum value of *FV* is 0.005 m/s and 0.004 m/s, respectively. Then, it decreases slowly within 9 m ahead of the tunnel face until reaching zero. The value of *FV* at the tunnel vault or floor is bigger than that at the tunnel face center. Near the excavation face, the *FV* order is $U_{Y=1.95} = U_{Y=-1.95} > U_{Y=0} > U_{Y=3.9} = U_{Y=-3.9}$.

In terms of water inflow, we acquire the water inflow rate of 670.37 m³/h at the excavation face and the total water inflow rate of 1813.16 m³/h by integrating the *FV* over the area.

### 3.4. Tunnel Is 20 m above the Center of the Two Intersecting Faults

Figure 8 reveals *PP* and *FV* contours on various sections ($XZ_{Y=-20}$, $XY_{Z=0}$, $YZ_{X=0}$) while the tunnel is excavated to 20 m above the two intersecting faults (Y = −20).

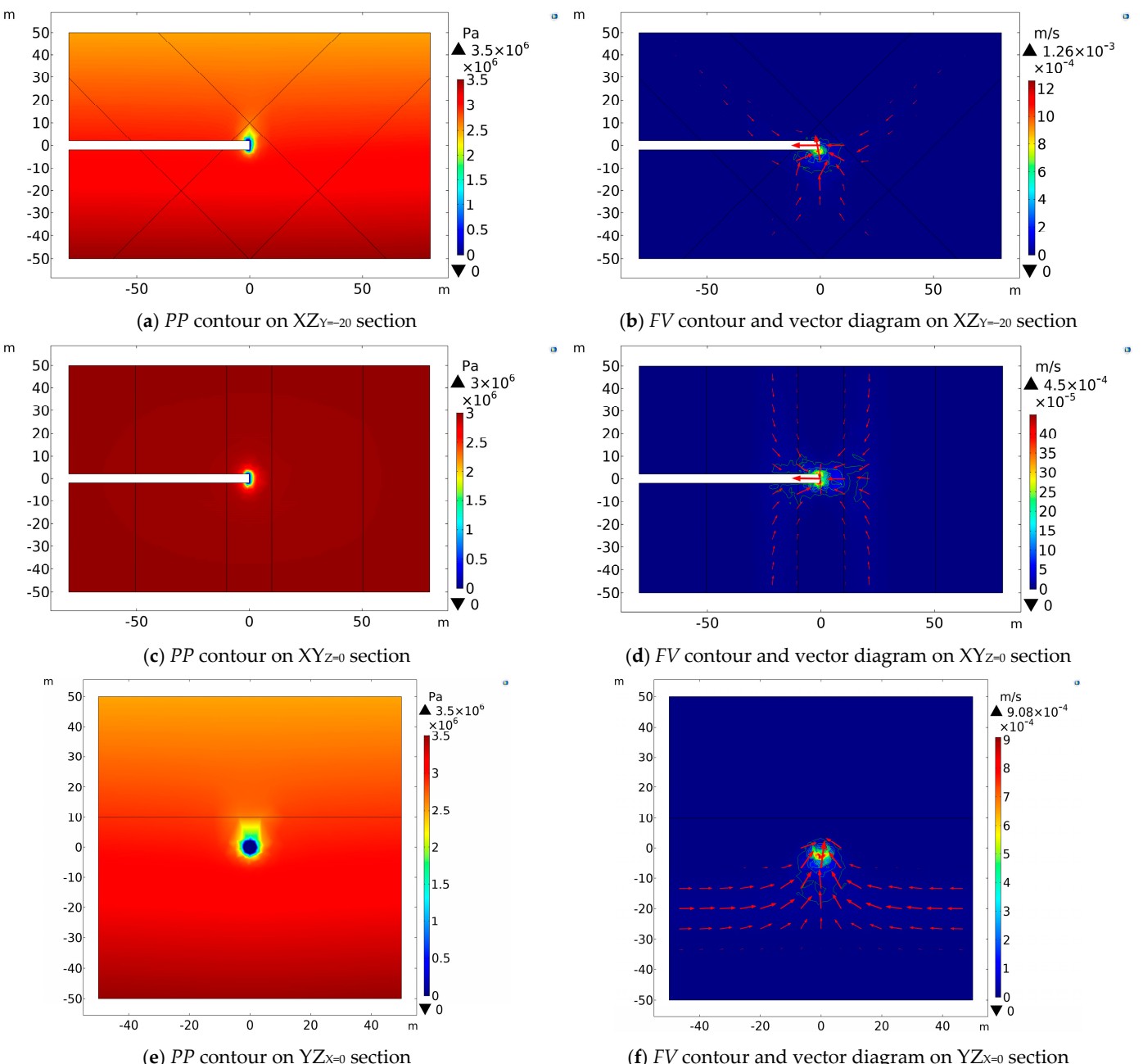

(**a**) *PP* contour on XZ$_{Y=-20}$ section

(**b**) *FV* contour and vector diagram on XZ$_{Y=-20}$ section

(**c**) *PP* contour on XY$_{z=0}$ section

(**d**) *FV* contour and vector diagram on XY$_{z=0}$ section

(**e**) *PP* contour on YZ$_{X=0}$ section

(**f**) *FV* contour and vector diagram on YZ$_{X=0}$ section

**Figure 8.** Simulation results for the case Y = −20 m.

As shown in the *PP* contours, a low-pressure region is seen at the excavation face and the tunnel wall within 1.0 m rearwards and increases. Specifically, the value of *PP* is 0 at the excavation face and then quickly increases with the increasing of *D*. On the cross-section of XZ$_{Y=0}$, the *PP* contours are distributed elliptically symmetrically along the *x*-axis, and the ellipse's major axis is parallel to the *z*-axis. On the vertical downward section, pore pressure gradually increases with the burial depth and is displayed in a funnel-shape near the excavation face. As shown in the *FV* contours, groundwater mainly flows from the overlapped intersection zone and the damage zone of the faults to the tunnel, with a maximum value of $1.26 \times 10^{-3}$ m/s. The *FV* contour appears symmetrically with respect to the *x*-axis on the section of XZY = 0 and the *y*-axis on the section of YZ$_{X=0}$. The value of *FV* underneath the tunnel floor is expressively bigger than that above the tunnel vault.

Five measuring lines were selected 50 m in front of the excavation face to monitor and explore the variation law of *PP* and *FV* as listed in Table 2 and Figure 9.

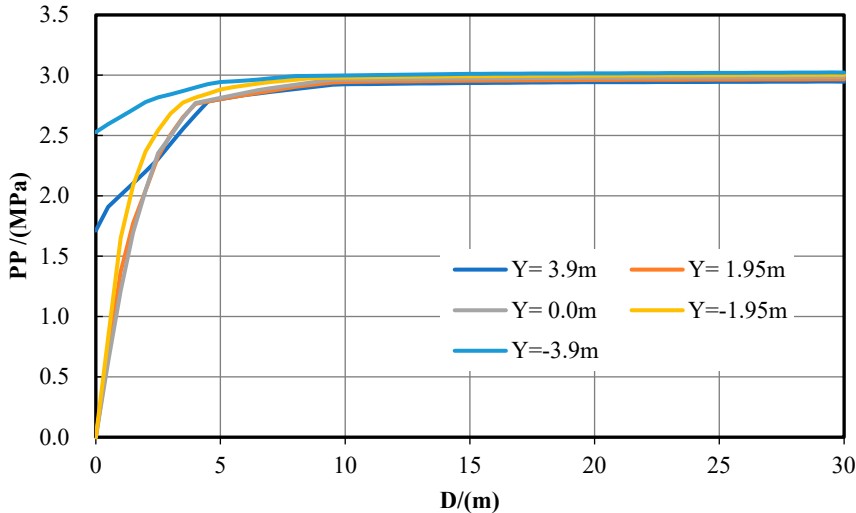

(**a**) *PP* as a function of distance away from the tunnel face(*D*)

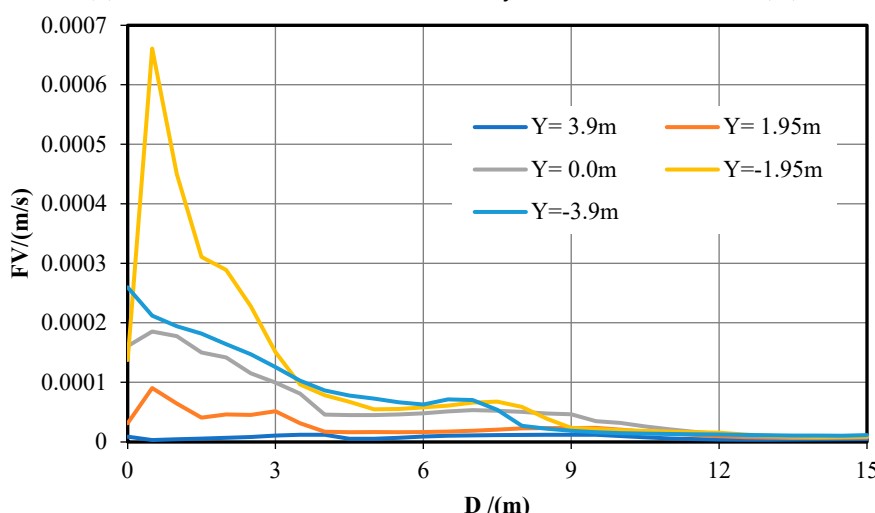

(**b**) *FV* as a function of distance away from the tunnel face(*D*)

**Figure 9.** Pore pressure and flow velocity within 30 m ahead of the tunnel face as a function of distance away from the tunnel face (Y = −20 m).

Figure 9a shows that, on the Y = 3.9 m section, the value of PP at the excavation face is about 2.53 MPa, and then gradually grows with the increase in D within 10 m ahead and eventually stabilizes at 3 MPa. On the Y = −3.9 m section, the value of *PP* at the excavation face is 1.71 MPa, and gradually grows within 5 m ahead and eventually stabilizes at 3 MPa. On the sections of Y = 1.95 m, 0, and −1.95 m, *PP* at the excavation face is 0 and increases rapidly within 5 m ahead, then it grows tardily and stabilizes at the maximum value of 3 MPa finally in the 5 m~15 m range. The *PP* value on sections of Y = ±3.9 m is significantly bigger than that on sections of Y = ±1.95 m and 0 within 5 m ahead.

As shown in Figure 9b, on the section of Y = 3.9 m, the value of *FV* is about 0. When on the sections of Y = 1.95 m, 0, −1.95 m, and −3.9 m, the maximum value of *FV* is $9.06 \times 10^{-5}$ m/s, $1.85 \times 10^{-4}$ m/s, $6.62 \times 10^{-4}$ m/s, and $2.60 \times 10^{-4}$ m/s, respectively. Flow velocities are gradually reduced within a 9 m range of the excavation face until reaching zero. In the range of 0~5 m of the host rock zone, the flow velocities below the tunnel floor are bigger than that at the tunnel vault and above, and the flow velocities at the tunnel vault or floor are bigger than that at the tunnel face and the surrounding areas. The magnitude order of the velocities is $U_{Y=-1.95} > U_{Y=-3.9} > U_{Y=0} > U_{Y=1.95} > U_{Y=3.9}$.

In terms of water inflow, we acquire the water inflow rate of 12.04 m$^3$/h at the excavation face and the total water inflow rate of 34.51 m$^3$/h by integrating the *FV* over the area.

### 3.5. Tunnel Is 40 m above the Center of the Two Intersecting Faults

Figure 10 reveals *PP* contours and *FV* contours on various sections (XZ$_{Y = -40}$, XY$_{Z = 0}$, YZ$_{X = 0}$) while the tunnel is excavated to 40 m above these two intersecting faults (Y = $-40$).

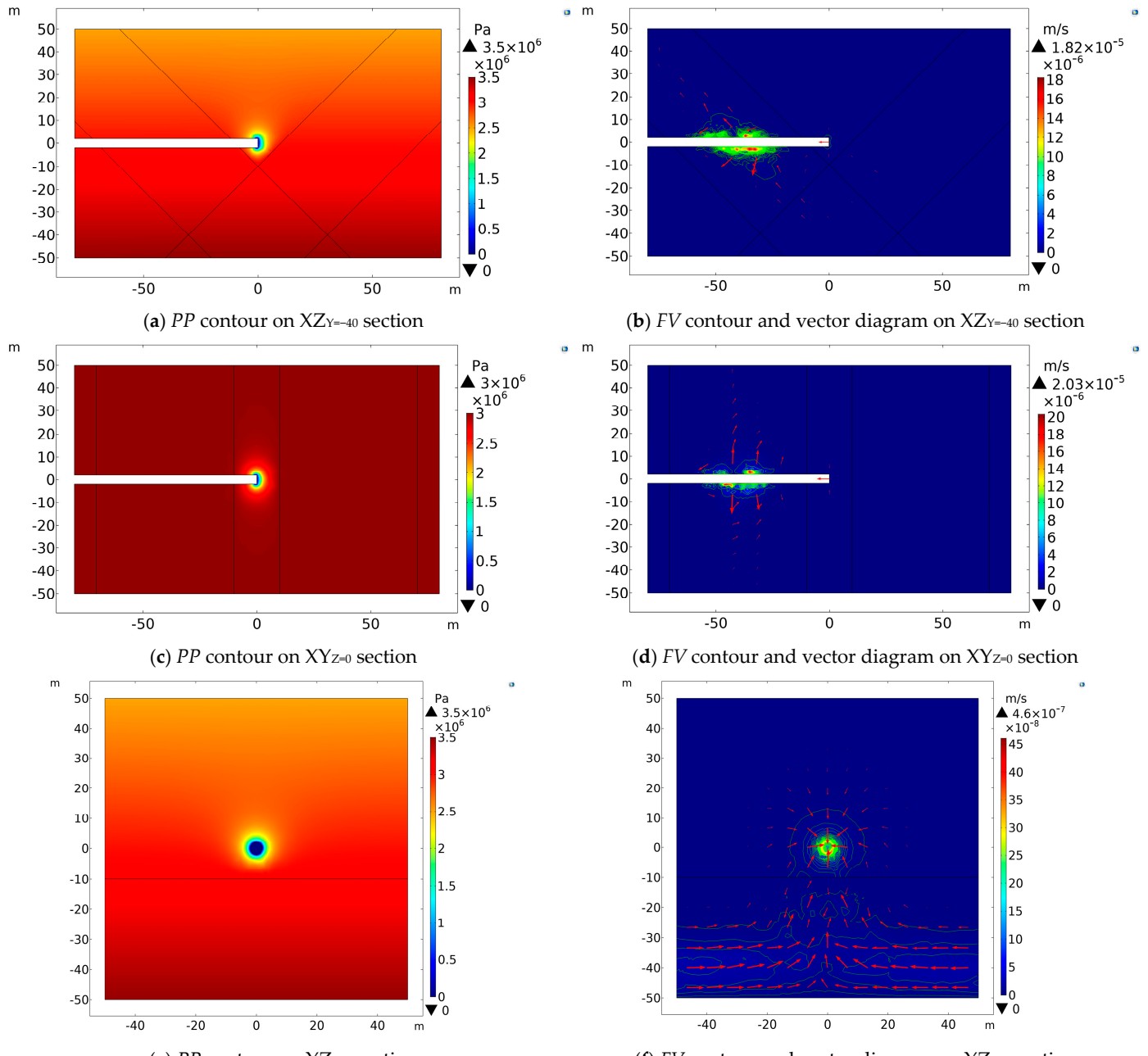

(**a**) *PP* contour on XZ$_{Y=-40}$ section

(**b**) *FV* contour and vector diagram on XZ$_{Y=-40}$ section

(**c**) *PP* contour on XY$_{Z=0}$ section

(**d**) *FV* contour and vector diagram on XY$_{Z=0}$ section

(**e**) *PP* contour on YZ$_{X=0}$ section

(**f**) *FV* contour and vector diagram on YZ$_{X=0}$ section

**Figure 10.** Simulation results for the case Y = $-40$ m.

The *PP* contours in Figure 10a,c,e all show a low-pressure region near the excavation face. Specifically, the value of *PP* is 0 at the excavation face and quickly increases outward with the increase in *D*. On the XZ$_{Y = -40}$ section, the *PP* contours are symmetrically elliptical with respect to the *x*-axis, and the central axis is parallel to the *z*-axis. On the YZ$_{X = 0}$ section, pore pressure increases with the burial depth and is presented in a funnel shape

near the excavation face. The *FV* contours reveal that groundwater mainly flows from the overlapped intersection zone and the damage zone of faults to the tunnel with a small *FV* and the maximum value is just about $2.03 \times 10^{-5}$ m/s. Five measuring lines were placed 50 m in front of the excavation face to monitor and explore the variation law of *PP* and *FV* as listed in Table 2 and Figure 11.

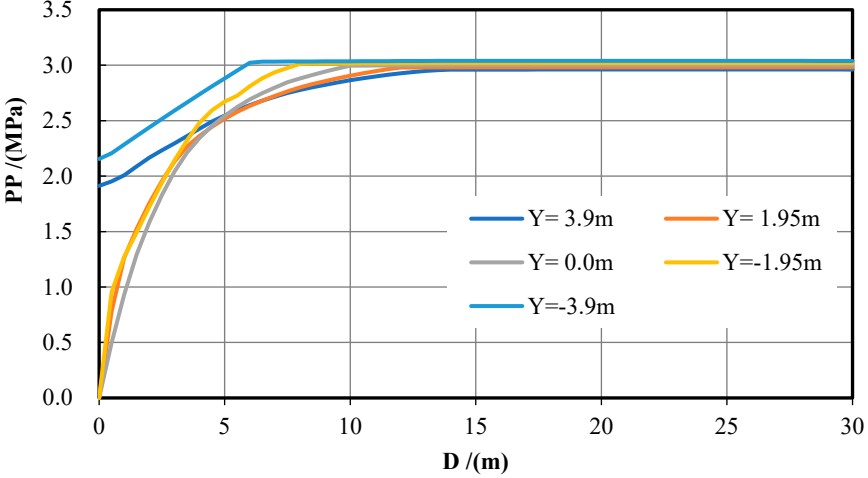

(**a**) *PP* as a function of distance away from the tunnel face (*D*)

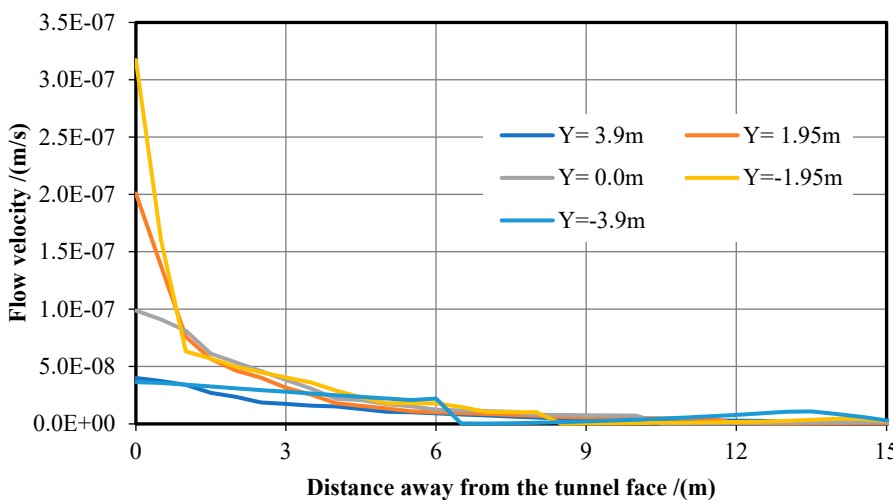

(**b**) *FV* as a function of distance away from the tunnel face (*D*)

**Figure 11.** Pore pressure and flow velocity within 30 m ahead of the tunnel face as a function of distance away from the tunnel face (Y = −40 m).

Figure 11a shows that, on the sections of Y = 3.9 m and Y = −3.9 m, the values of PP increase gradually from 1.91 MPa to 2.15 MPa at the excavation face within 15 m and 6 m on the two sections, respectively, and finally stabilize approximately at 3 MPa. On the sections of Y = 1.95 m, 0, and −1.95 m, the value of *PP* at the excavation face is 0, it increases rapidly first in the range of 5 m~15 m and then slowly afterwards, and finally stabilizes at 3 MPa. Within 5 m ahead, the *PP* value on the Y = ±3.9 m section is significantly bigger than on the sections of Y = ±1.95 m and Y = 0. Figure 11b shows that, on the sections of Y = 3.9 m and Y = −3.9 m, *FV* decreases rapidly from the maximum value of $4.00 \times 10^{-8}$ m/s and $3.64 \times 10^{-8}$ m/s. On the section of Y = 0, *FV* decreases slowly from the maximum value of $9.88 \times 10^{-8}$ m/s until it approaches 0. Within 9 m in front of the excavation face, *FV* decreases gradually first and then slowly until it approaches zero. In 0~3 m of the host rock zone, *FV* at and below the tunnel floor is much higher than that at the tunnel vault

and above, and *FV* at the tunnel vault or floor is bigger than that at the tunnel face and the surrounding areas. Their magnitudes are in the order of $U_{Y=-1.95} > U_{Y=1.95} > U_{Y=0} > U_{Y=-3.9} > U_{Y=3.9}$.

In terms of water inflow, we acquire the inflow rate of 0.0061 $m^3$/h at the excavation face and the total inflow rate of 0.016 $m^3$/h by integrating the *FV* over the area.

## 4. Discussion

In the above section, we presented the results of the pore pressure (*PP*), the flow velocity (*FV*), and water inflow rate at various vertical relative locations between the excavation face and the center of the two overlapped intersecting faults. In the below section, we combine these results and obtain the influence of the relative location of the intersecting faults on water inflow into the tunnel. Figure 12 shows the pore pressure curves from the survey lines in the center of the excavation face.

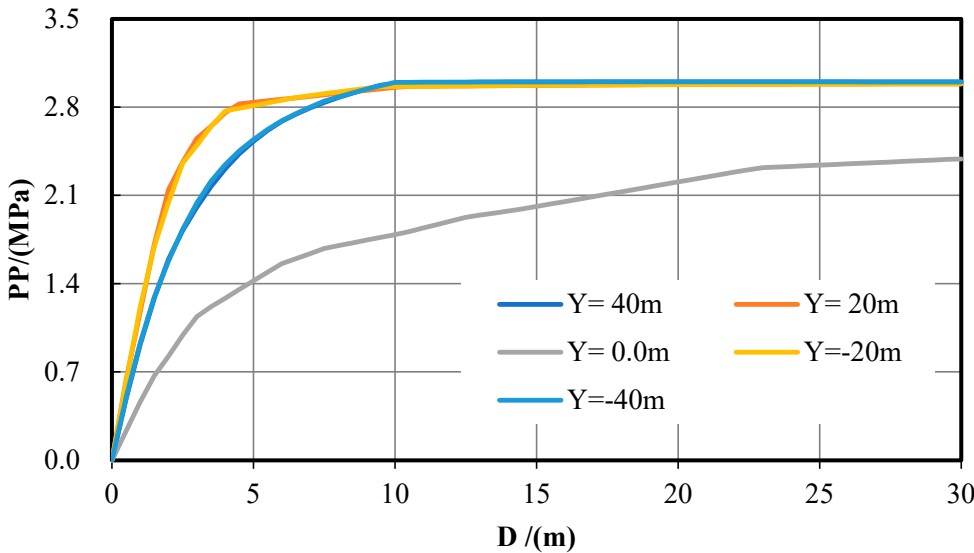

**Figure 12.** *PP* as a function of *D* with various relative positions.

It can be obtained from Figure 12 that pore pressures at the Y = ±40 m and Y = ±20 m sections are very similar. Within 10 m range ahead of the tunnel face, fluid pressure flows are $P_{Y=20} = P_{Y=-20} > P_{Y=40} = P_{Y=-40} > P_{Y=0}$. In summary, when the fault intersection is in Y = ±40 and Y = ±20, the surrounding rock near the tunnel face has low permeability, which has little effect on the water pressure around the excavation face. Due to the stress release during the tunnel excavation progress, a low-pressure region is formed near the excavation face, which has a more significant influence on the distribution of pore pressure in the tunnel-surrounding rock than the influence of the two overlapped intersecting faults.

We select the flow velocity at each point from the survey lines in the center of the excavation face (Y = 0 m) to obtain the flow velocity curves, as shown in Figure 13. The flow velocity magnitude follows the order of $U_{Y=0} > U_{Y=20} = U_{Y=-20} > U_{Y=40} = U_{Y=-40}$. When the tunnel approaches the fault intersection, the flow velocity increases by orders of magnitude, reflecting that the distance to the fault intersection significantly impacts the fluid flow velocity in the tunnel-surrounding rock. The water inflow rate around the excavation face and the total water inflow rate into the tunnel are shown in Table 3 and Figure 14. It has been found that when the fault intersection center position is at Y = 0, the inflow rate value is the largest, and when the fault intersection center position is at Y = ±40 m and Y = ±20 m, the water inflow rates are the same. It shows that the relative vertical location of the intersecting faults has little effect on the flow field.

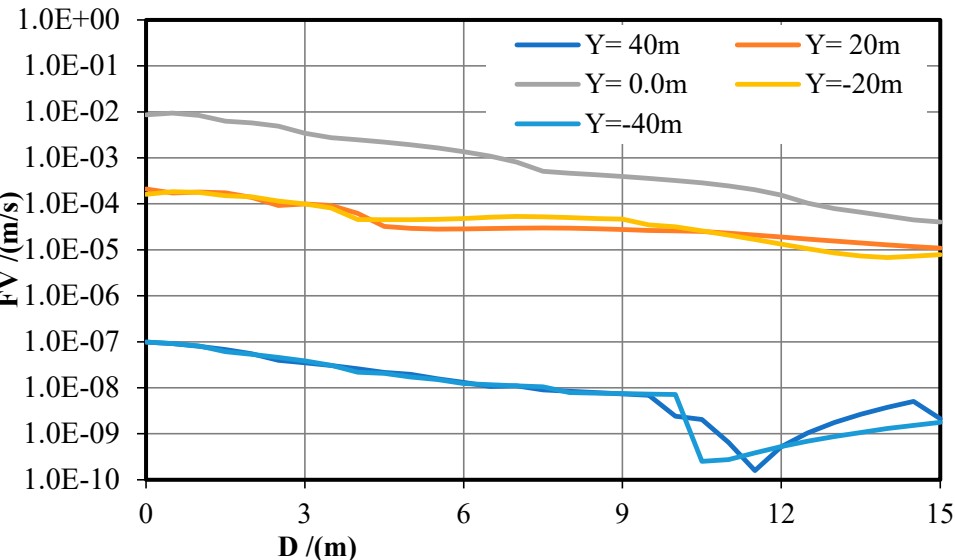

**Figure 13.** *FV* as a function of *D* with various relative positions.

**Table 3.** Water inflow rates with various relative locations of the intersecting faults.

| Location of the Fault Intersection | Y = 40 m | Y = 20 m | Y = 0 m | Y = −20 m | Y = −40 m |
|---|---|---|---|---|---|
| Flow rate at the excavation face ($m^3$/h) | 0.005 | 12.5 | 670.4 | 12.0 | 0.006 |
| Flow rate at the tunnel perimeter ($m^3$/h) | 0.010 | 23.2 | 1142.8 | 22.5 | 0.010 |
| Total flow rate ($m^3$/h) | 0.015 | 35.7 | 1813.2 | 34.5 | 0.016 |

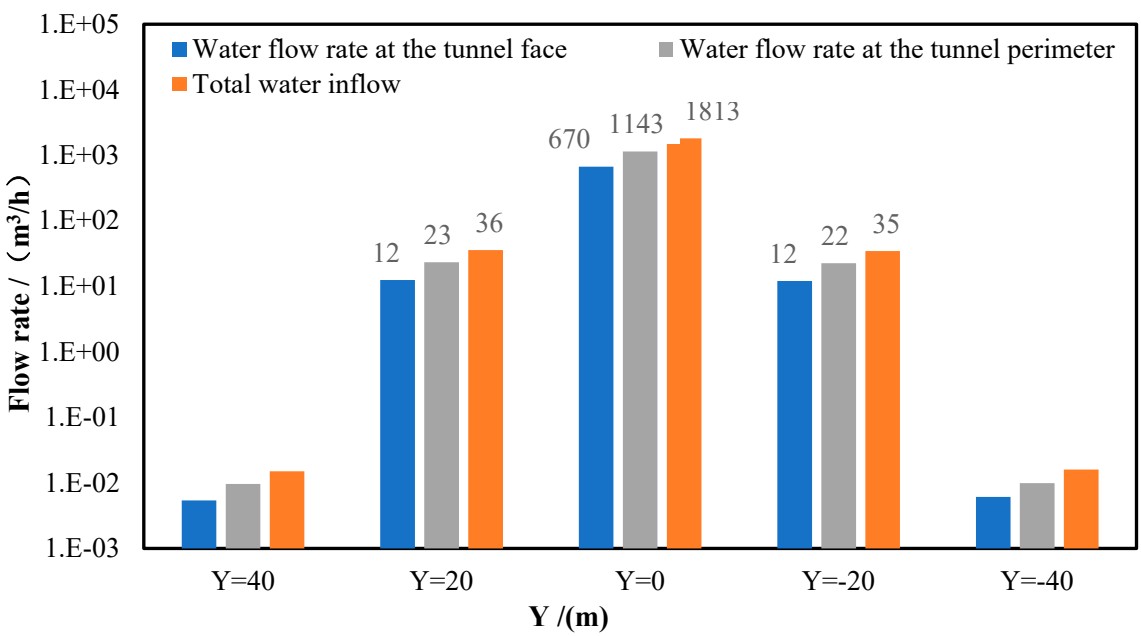

**Figure 14.** Bar chart of water inflow rates with different relative locations of the fault intersection (Y).

## 5. Conclusions

In this work, we developed an improved Darcy-Brinkman flow model to explore the behavior of an underground water flow and inrush into a tunnel crossing two overlapped intersecting faults. We evaluated the influence of the relative vertical position between the tunnel axis and the center of the two intersecting faults on pore pressure, flow velocity,

and inflow rate evolution law. Results show that the location of the two intersecting faults has a negligible effect on pore pressure around the tunnel-surrounding rock. Specifically, the pore pressure increases quickly initially and slowly with the increase in the distance away from the excavation face. The pore pressure near the excavation face is smaller than outward in the *y*-axis direction; it decreases as Y increases away from the excavation face. From the analysis results of flow velocity, it is evident that the most significant flow velocity values appear near the excavation face. Then, it declines as the distance from the excavation face increases until it stabilizes at a small value, as in the host rock zone. For the water inflow rate, we find that when the tunnel's excavation face is just in the center of the overlapped intersecting faults, the inflow rate is the biggest, while the rates are the same at the Y = ±40 m and Y = ±20 m sections. Therefore, the flow rate change around the tunnel mainly depends on the absolute value of the distance between the fault's intersection center and the tunnel axis and is independent of their relative vertical positions. Based on the theory of "Three-district zoning of faults", numerical simulation can be used to study the water inrush behavior into tunnels and provides an effective method for simulating groundwater flow in tunnel-surrounding rock while the tunnel excavation face passes through two overlapped intersecting faults. This method provides a new idea for simulation and a helpful reference for predicting water inflow rates in underground projects.

**Author Contributions:** Conceptualization, L.W. and J.W.; methodology, J.W.; software, J.W. and M.S.; validation, Y.H. and M.S.; data curation, M.S. and Y.H.; writing—original draft preparation, J.W.; writing—review and editing, Y.L.; funding acquisition, J.W., Y.L., and M.S. All authors have read and agreed to the published version of the manuscript.

**Funding:** This research was funded by the National Natural Science Foundation of China,41907259; the Hubei Natural Science Foundation of Hubei Province of China, 2022CFB948; the Foundation of Engineering Research Center of Rock-Soil Drilling & Excavation and Protection, 202208 and 202215.

**Institutional Review Board Statement:** Not applicable.

**Informed Consent Statement:** Not applicable.

**Data Availability Statement:** Not applicable.

**Acknowledgments:** Thanks for the help of Zhenhao Xu and Xintong Wang. They offered advice on the composition and technical support on the numerical modeling of the article.

**Conflicts of Interest:** The authors declare no conflict of interest.

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
