# Peer review of "Numerical Investigation of Water Inflow Characteristics in a Deep-Buried Tunnel Crossing Two Overlapped Intersecting Faults"

_water, doi:10.3390/w15030479_

Round 1

Reviewer 1 Report

This paper examines water inflow to a deep tunnel in the vicinity of intersecting faults in bedrock.  The model analysis should be of interest to geotechnical engineers concerned with tunnel construction and dewatering.  The mathematics used seem appropriate, and the results are interesting, but in my opinion not surprising.

The paper has a weakness in that the authors have apparently done the analysis using fixed values for all parameters and do not consider the geologic heterogeneity that would always be present in a geological system such as they describe.  Fixed values are used for important parameters such as permeability and porosity.  In a real system these parameters would likely be both uncertain and spatially variable.  The only parameter varied in the analyses is the location of the tunnel relative to the faults.  Results for tunnel inflow and pore pressure are given as fixed numbers rather than ranges.  An uncertainty analysis would make this a much stronger paper.

The paper also lacks a good description of the conceptual model being simulated.  Any modeling exercise depends on a clear conceptual model showing the problem to be solved and appropriate boundary conditions.  The paper attempts to describe the conceptual model in words in lines 101-123, but a good diagram would make this description much easier to follow.   Other hints about the conceptual model are buried in the text between lines 139 and 142, which states that the “rest supported tunnel wall” (unclear what this refers to) and bottom boundary are impermeable. The current figure 1 is not adequate to do this.  Apparently (as noted in line 418) there will be supplementary materials that include a 3-D visualization of the simulated system, but this was not included in the review materials.  I strongly recommend that the authors replace figure 1 with a 3-D perspective image of the problem domain so that the reader can understand the orientation of the tunnel and the intersecting faults.

The paper should clearly state that this is a steady-state model.  Transient simulations, which might be more realistic, would almost certainly produce different results.

Please elaborate on how the COMSOL modeling tools were used. 

I found all of the heat-map output figures (such as figure 3) difficult to read either in printed format or enlarged on my computer screen.  In particular the vector arrows are too small to see and the contour lines, if that’s what they are, are very faint.  I do not understand the units on the color scale, which seem to read (for example fig 3b) 2.95x10-3x10-4.   Please indicate units on all diagrams.

Some specific comments:

Line 99 (and elsewhere):  Please check English usage.  “…high permeability because of with no minerals filled.”  Etc.

Line 108: States the permeability of the host rock is 10-16 m2.  How was this value chosen?  What rock type does it represent?  What is the uncertainty in this value?

Line 109:  State the permeability of the fault core is 10-11 m2.  Same questions as above.

Line 109:  What do you mean by “variation in permeability… obeys a Gaussian function”?  How was this incorporated into the model?

Line 114:  States that the porosity of the fault core is 0.5.  This seems extraordinarily high to me; it implies that the fault core is 50% open space, which seems impossible.  Is there field evidence for this?

Author Response

We would like to thank you for your review of our manuscript entitled “Numerical Investigation of Water Inflow Characteristics in a deep buried tunnel crossing two overlapped intersecting faults” with the manuscript number water-2154708. Those advices and suggestions are significant valuable and helpful for improving the quality of our manuscript. We have considered the comments and checked the manuscript carefully, then made corrections which we hope meet the publishing requirements. Revised portion are marked in red in the manuscript. In addition, due to the grammatical errors in the manuscript, we didn’t mark them all in the revised manuscript. The main corrections in this manuscript and the itemized responds to your comments are as flowing:

Comment 1: The paper attempts to describe the conceptual model in words in lines 101-123, but a good diagram would make this description much easier to follow. Other hints about the conceptual model are buried in the text between lines 139 and 142, which states that the “rest supported tunnel wall” (unclear what this refers to) and bottom boundary are impermeable. The current figure 1 is not adequate to do this. I strongly recommend that the authors replace figure 1 with a 3-D perspective image of the problem domain.

Response 1: Thanks for your suggestion. In addition to the text description, the numerical simulation model of a tunnel crossing two overlapped intersecting faults can be found in Figure 1. Some impermeable boundaries have been added in Figure 1. We have retained the 3-D perspective image of the study area in Figure 1, because the 3D model may make the fault look more complex. At the same time, we have revised Figure 1 and specified some boundary conditions in section 2.2 so that the reader can understand the orientation of the tunnel and the intersecting faults. Revised portion are highlighted by using red text.

Comment 2: The paper should clearly state that this is a steady-state model. The paper should clearly state that this is a steady-state model.

Response 2: Thanks for your suggestion. We have state that this is a steady-state model clearly in the revised paper. Revised portion are highlighted by using red text.

Comment 3: Please elaborate on how the COMSOL modeling tools were used.

Response 3: We have added some details on how the COMSOL modeling tools were used in the revised paper. Revised portion are highlighted by using red text. The revised parts can be found in Section 2.2 Numerical simulations.

Comment 4: I found all of the heat-map output figures (such as figure 2) difficult to read either in printed format or enlarged on my computer screen. In particular the vector arrows are too small to see and the contour lines, if that’s what they are, are very faint. I do not understand the units on the color scale, which seem to read (for example fig 2b) 2.95x10-3-x10-4. Please indicate units on all diagrams.

Response 4: Thanks a lot for your kind review. We have revised the vector arrows and the contour lines for the heat-map output figures in the revised paper, and the units have been adjusted (for example, it should be 2.95×10-3 for fig 2b).

Comment 5: Line 99 (and elsewhere): Please check English usage.

Response 5: We have revised the grammatical errors in Line 99 to make it more readable. We are very sorry for our grammatical mistakes. We have made corrections according to the reviewer’s comments and have polished the manuscript with a professional assistance.

Comment 6: Line 108: States the permeability of the host rock is 10-16 m2. How was this value chosen? What rock type does it represent?

Response 6: In this work, the host rock represents limestone and dolomite. Thus, the permeability is chosen as 10-16 m2. We have added it in the revised paper. Revised portion are highlighted by using red text.

Comment 7: Line 109: State the permeability of the fault core is 10-11 m2. How was this value chosen?

Response 7: Thanks for your kind review. We obtain the permeability value according to a field water pressure test carried out in the vicinity of F61 fault of a water diversion tunnel engineering as shown in our earlier published article. The permeability coefficients of the core zone and damage zone were calculated, where the permeability of the host rock is 10-16 m2 and the permeability of the fault core is 10-11 m2. We also fitted the variation law of permeability between the host rock and the fault core by Gaussian function fund.

Comment 8: Line 109: What do you mean by “variation in permeability… obeys a Gaussian function”? How was this incorporated into the model?

Response 8: Thanks for this comment. In this work, the permeability of host rock zone is 10-16m2 and the maximum permeability of the fault core zone is 10-11m2 for the numerical model. There is a damage zone between the host rock zone and the fault core zone, and the permeability from the host rock to the fault core follows the Gaussian function. The permeability change can be achieved in the software COMSOL.

Comment 9: Line 114: States that the porosity of the fault core is 0.5. This seems extraordinarily high to me; it implies that the fault core is 50% open space, which seems impossible. Is there field evidence for this?

Response 9: Thank you for your comments. When the filling degree in the vicinity of the fault core is very small, we considered it a typical water conduction structure. In order to facilitate calculation, we also simplify the fault core area into a homogeneous pore structure, and the porosity value of 0.5 is possible. As previously mentioned in comment 6 and comment 7, we had carried out a field water pressure test in the vicinity of a fault rely on a water diversion tunnel project, and the test data also well proves this.

Reviewer 2 Report

The comments to the authors:

1 - Introduction can be improved by other references (18 references in the article) (why AFH mentioned in the introduction);

2 - All figures must be repeated clearly;

3 - Methods and methods must be improved including more details about simulation software;

4 - Possibility to perform 3D model;

5 - English editiong certificate is needed.

Author Response

We would like to thank you for your review of our manuscript entitled “Numerical Investigation of Water Inflow Characteristics in a deep buried tunnel crossing two overlapped intersecting faults” with the manuscript number water-2154708. Those advices and suggestions are significant valuable and helpful for improving the quality of our manuscript. We have considered the comments and checked the manuscript carefully, then made corrections which we hope meet the publishing requirements. Revised portion are marked in red in the manuscript. In addition, due to the grammatical errors in the manuscript, we didn’t mark them all in the revised manuscript. The main corrections in this manuscript and the itemized responds to your comments are as flowing:

Comment 1: Introduction can be improved by other references (18 references in the article);

Response 1: Thanks for your careful review. We have added more related references in and revised the Introduction. Revised portion are highlighted by using red text.

Comment 2: All figures must be repeated clearly;

Response 2: Thanks for your comment. We have provided clear figures in the revised paper.

Comment 3: Methods and methods must be improved including more details about simulation software;

Response 3: We have improved the methods and added some details for the COMSOL modeling tools in the revised paper. Revised portion are highlighted by using red text.

Comment 4: Possibility to perform 3D model;

Response 4: Thanks for your kind review. This paper established an improved Darcy-Brinkman numerical model for a tunnel crossing the intersecting faults for numerical investigation of water inflow characteristics in a deep buried tunnel. At the same time, we are currently conducting the 3D model numerical study on the water inflow characteristics of deep-buried tunnels which will be submitted in the near future.

Comment 5: English editing certificate is needed.

Response 5: Thank you for pointing this out. We have revised some grammatical errors carefully. We also have made corrections according to the reviewer’s comments and have polished the manuscript with a professional assistance. Please refer to the attachment for English editing certificate.